# Adversarial Robustness Unhardening via Backdoor Attacks in Federated Learning

**Taejin Kim**[1]    **Jiarui Li**[2]    **Shubhranshu Singh**[2]    **Nikhil Madaan**[2]    **Carlee Joe-Wong**[2]

[1]CACI Intl. Inc.
[2]Carnegie Mellon University
[1]`taejin.kim@caci.com`
[2]`{jiaruil3,shubhran,nmadaan,cjoewong}@andrew.cmu.edu`

## Abstract

In today's data-driven landscape, the delicate equilibrium between safeguarding user privacy and unleashing data's potential stands as a paramount concern. Federated learning, which enables collaborative model training without necessitating data sharing, has emerged as a privacy-centric solution. This distributed approach brings forth security challenges, notably poisoning and backdoor attacks where malicious entities inject corrupted data. Our research, initially spurred by test-time evasion attacks, investigates the intersection of adversarial training and backdoor attacks within federated learning, introducing Adversarial Robustness Unhardening (ARU). ARU is employed by a subset of adversaries to intentionally undermine model robustness during federated training, rendering models susceptible to a broader range of evasion attacks. We present extensive empirical experiments evaluating ARU's impact on adversarial training and existing robust aggregation defenses against poisoning and backdoor attacks. Our findings inform strategies for enhancing ARU to counter current defensive measures and highlight the limitations of existing defenses, offering insights into bolstering defenses against ARU.

## 1   Introduction

Federated learning has emerged as a promising distributed training paradigm [1, 2] to address privacy concerns linked with the escalating growth and utilization of sensitive data generated and collected by modern computing devices, such as smartphones and Internet of Things (IoT) sensors. It allows multiple users to collaboratively train a machine learning model without the necessity of sharing their individual data. Instead, during the training phase, participants autonomously update the model using their respective data while safeguarding data privacy. The proliferation of federated learning, alongside machine learning in general, has facilitated advancements in numerous applications. However, it has also ushered in a wave of attacks on learning algorithms. Evasion attacks [3, 4], for instance, aim to manipulate inputs to trained models in ways imperceptible to human users but capable of altering the model's output during testing. For instance, making slight alterations to a stop sign might result in its misclassification as a speed limit sign [5]. Alternatively, backdoor and poisoning attacks have adversarial clients participating in model training, sending manipulated weight information during aggregation to negatively impact the trained global model [6].

**Threat Model.** Existing defenses against evasion attacks in federated learning generally utilize adversarial training [7, 8, 9, 10], where clients generate adversarial inputs and incorporate them into the training process, which has been shown to be an effective and reliable defense method against evasion attacks [4]. However, such defenses only consider attacks generated and deployed during the testing phase of the federated learning model. In this paper, we present a *novel train-time backdoor adversary that complements such test-time evasion attack adversaries* by discreetly interfering with

Published at NeurIPS 2023 Workshop on Backdoors in Deep Learning: The Good, the Bad, and the Ugly.

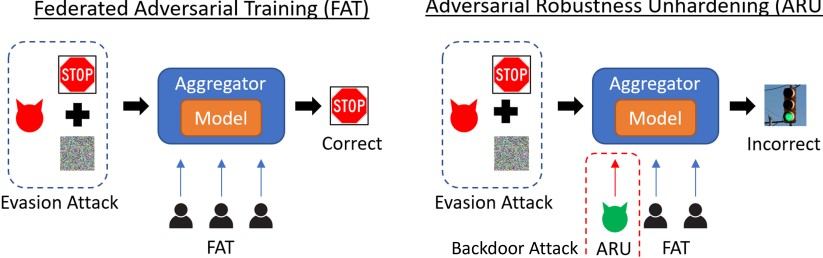

Figure 1: With federated adversarial training, the clients jointly train a model robust against evasion attacks (left). However, when a small subset of train-time clients perform the ARU backdoor attack, the jointly trained model performs poorly against evasion attacks (right).

the federated adversarial training process. As seen in Figure 1, the interfering adversary successfully reduces the classification accuracy against evasion attacks (i.e., robustness) of the trained model, while leaving performance against benign, unaltered inputs high to avoid detection by other clients.

**Challenges and Contributions.** The primary challenge in implementing the proposed ARU attack lies in the fact that the attacking clients constitute only a small subset of the total client pool. Consequently, any attack strategy must operate within the constraints of the limited data and computational resources available to these attacking clients. Future challenges include devising methods to surmount robust aggregation defenses against poisoning and backdoor attacks. In Section 5, we delve into potential directions for addressing this challenge and enhancing the resilience of federated learning systems. In this paper, our main contributions are the following:

- We are the first, to the best of our knowledge, to characterize and analyze the *coordination between train-time (backdoor) and test-time adversaries for enhancing the success rate of evasion attacks* via Adversarial Robustness Unhardening (ARU). Although existing backdoor methods have a similar concept, instead of embedding a backdoor for a narrow subset of inputs, ARU enhances the performance of all evasion attacks.

- We propose a realistic attack scheme, ARU-Extract, given a small number of collaborating attacking clients, that obtains information necessary for ARU during the training phase.

- To assess the impact of ARU on the security of federated learning models, we conduct a thorough evaluation of existing defense mechanisms, namely robust aggregation schemes for federated learning. We include an extended discussion to enhance ARU against such defenses, as well as the weakness of such existing defenses.

The remainder of this paper is organized as follows. Section 2 contrasts ARU with related works. Section 3 introduces the federated adversarial training procedure, and evaluates its impact on test-time evasion attacks. Section 4 introduces the train-time ARU attack, its impact on the robustness of trained models, and realistic deployment methods given adversarial constraints. In Section 5, we test ARU against existing robust aggregation defenses, and perform an extended discussion on the next steps of this research. We conclude in Section 6.

## 2 Related Works

**Backdoor and Poisoning Attacks.** Within the training phase of federated learning systems, malicious clients have the potential to engage in backdoor or poisoning (Sybil) attacks, where colluding clients send manipulated weight information during the aggregation phase with the intent of degrading the performance of the global model, either generally or for specific sub-tasks [6]. To enhance the persistence and efficacy of these attacks, [11] introduces a distributed attack strategy that employs localized triggers to poison individual attacker models while collectively exploiting the shared model. Additionally, [12] presents edge-case backdoor attacks that target prediction sub-tasks unlikely to be encountered in the training or test data sets, yet still represent plausible real-world scenarios.

**Defense Methods.** Robust aggregation schemes have been proposed such as Krum, Bulyan, and trimmed-mean as a defense mechanism against backdoor and poisoning attacks [13, 14, 15, 16]. While these schemes prove highly effective in countering simple or non-intelligent attacks, they

become vulnerable to exploitation by a byzantine attacker who possesses knowledge about the specific aggregation scheme being employed [17]. Further, the authors of [18] propose a novel attack on a broad range of robust aggregation schemes, where they tailor perturbations to the sensitivity curve of the aggregator. Hence, to safeguard the system against such attacks, [19] introduces a novel approach to counter malicious attacks by avoiding the use of a fixed aggregation scheme. Alternate defense methods against backdoor and poisoning attacks have also been proposed such as FLAME [20] that estimates the noise that needs to be injected into the global model to eliminate backdoors, combined with dynamic clustering and adaptive clipping techniques. The work in [21] adjusts the learning rate on the server based on the sign of client updates to avoid backdoor attacks.

## 3 Federated Adversarial Training

We first describe the target ARU attempts to undermine: federated adversarial training (FAT), which aims to train a global model through federated learning that is robust against evasion attacks. In essence, adversarial training includes evasion attack data points within its training set to gain familiarity and robustness against future test-time attacks.

### 3.1 Crafting Evasion Attacks

A test-time adversary performs an evasion attack by altering the input $x$ to alter the model prediction for $x$. In this paper, we make the assumption that both adversarial training and evasion attacks are conducted using the well-known projected gradient descent (PGD) method [4]. The PGD method is one of the most popular and effective forms of evasion attacks [22]. Here, the adversary aiming to induce any incorrect classification label iteratively updates the current input $x^t$ as:

$$x^{t+1} = \Pi_{x+S} \left( x^t + \alpha \text{sgn}(\nabla_x L(h_\theta, x^t, y)) \right) \tag{1}$$

The input $x$ with correct label $y$ is perturbed along the gradient of the model loss function $L$ with model parameters $h_\theta$. The step size $\alpha$ is chosen to not be too small so that an effective perturbation can be quickly found, while not too large such that effective perturbations are not omitted. The perturbation to input $x$ is then projected ($\Pi_{x+S}$) to be within the perturbation budget $S$. The perturbation budget exists such that perturbations are not obvious to detection (e.g., a heavily perturbed image may be noticed by the human eye). This budget is most often a $l_2$ or $l_\infty$ norm-ball.

In this paper, we examine white-box attacks that are characterized by the attacker having full access to the internal architecture and parameters of the victim model, and is reasonable in a federated learning setting where many participating clients own a copy of the global model. This knowledge allows the attacker to craft evasion attacks with better precision, making them generally more effective. In contrast, black-box attacks operate under the assumption that the attacker has limited or no access to the victim model's internal structure, relying on input-output interactions or a substitute model to generate adversarial examples. Furthermore, all evasion attacks examined are *untargeted* attacks that aim to have inputs classified as any incorrect output.

### 3.2 The Adversarial Training Process

Defending against evasion attacks can be mathematically represented as a saddle point problem. In adversarial training, the primary objective is to train a model that minimizes the empirical risk associated with a classification task, even in the face of the adversary's introduction of input perturbations (e.g., through PGD as demonstrated in Equation 1) that maximize the loss at each data point [4]. The objective function of adversarial training is as follows:

$$\min_\theta \mathbb{E}_{(x,y)\sim\tilde{D}} \left[ \max_{\delta \in S} L_{\tilde{D}}(h_\theta, x + \delta, y) \right] \tag{2}$$

The perturbation $\delta$ added to the data $\tilde{D}$ is bounded within a budget $S$. In words, we desire to find model parameters $\theta$ that minimize the expected maximum loss when the input $x$ is perturbed by $\delta$. During FAT, each client incorporates adversarial examples into its local training data set by utilizing gradient information from its individual model [7]. These gradients are subsequently aggregated centrally following the standard Federated Learning Average (FedAvg) protocol [2].

The effects of federated adversarial training are examined and compared to FedAvg in Table 1. Data is split in a non-i.i.d. manner across 40 to 50 clients, as in [23], and is trained on the MobilenetV2

| Dataset | Metric | FedAvg | FAT | ARU | ARU-E |
|---------|--------|--------|-----|-----|-------|
| CIFAR10 | Test Acc. | 0.839 (0.04) | 0.781 (0.05) | 0.840 (0.04) | 0.667 (0.05) |
| | Adv. Acc. | 0.154 (0.08) | 0.662 (0.07) | 0.160 (0.07) | 0.068 (0.01) |
| CIFAR100 | Test Acc. | 0.539 (0.05) | 0.484 (0.05) | 0.538 (0.05) | 0.340 (0.06) |
| | Adv. Acc. | 0.169 (0.04) | 0.428 (0.05) | 0.161 (0.04) | 0.120 (0.04) |

Table 1: Test accuracy and robustness for different federated learning and ARU algorithms.

architecture for 200 rounds. Each client performs measurements on the global model with their local test data. Standard deviation values are represented in parentheses. Overall, FAT enhances the robustness of models (i.e., classification rate against evasion attacks, denoted as Adv. Acc.) compared to FedAvg. Next, we examine the robustness of the trained model in the presence of a train-time adversary abetting the test-time evasion attack while undermining FAT.

## 4 Threat Model: Adversarial Robustness Unhardening

The ARU adversaries aim to undermine FAT such that the trained model becomes much less robust against evasion attacks. **In essence, the adversaries are strategically embedding a significant backdoor into the model, effectively compromising its resilience against all forms of gradient-based adversarial evasion attacks**. Below, we first describe the model replacement method used by the ARU adversaries and discuss the subsequent adjustments made to enhance its practicality.

### 4.1 The Model Replacement Method

In the model replacement attack done by a single client $j$ out of $m$ clients as shown in [6], the attacker aims to substitute the legitimate global model of round $t$, $G^t$, with its own maliciously crafted model $R$, as seen in Equation 3. Here, $1/\gamma_i$ indicates the weighted contribution of the client $i$ during federated learning aggregation:

$$R = G^t + \sum_{i=1}^{m} \frac{1}{\gamma_i}(U_i^{t+1} - G^t) \tag{3}$$

To accomplish this, the attacker strategically ensures that its uploaded model $U_j^{t+1}$ survives the aggregation across all clients by boosting the difference between desired model $R$ and global model $G^t$ based on $\gamma_j$, as seen in Equation 4:

$$U_j^{t+1} = \gamma_j R - (\gamma_j - 1)G^t - \sum_{i=1}^{m-1} \frac{\gamma_j}{\gamma_i}(U_i^{t+1} - G^t) \tag{4}$$

$$\approx \gamma_j(R - G^t) + G^t \tag{5}$$

When $G_t$ is near convergence, the updates from benign participants are assumed to be close to zero ($U_i^{t+1} \approx G^t$), and are omitted in Equation 5. The work in [6] discusses how adversary $j$ can estimate $\gamma_j$ during the training procedure, if not known ahead of time. Furthermore, model replacement can be done jointly by multiple ARU clients, where assuming that contribution $\gamma_j$ is equal amongst all adversaries, they can alter their boosting rate to $\frac{\gamma}{N}$, where $N$ is the number of ARU clients.

In Table 1, a single adversary performs ARU by model replacement using a non-robust model trained by FedAvg. The ARU attack showcases a significant decrease in the robustness from FAT. However, it is an unreasonable to assume that the adversary owns the non-robust model used for replacement ahead of time. Thus, we next examine how a small number of collaborating clients can *extract* a non-robust model from the robust global model during the training process.

### 4.2 ARU-Extract: Extracting the Non-Robust Model During Training

While clients attempting to perform ARU do not reasonably have access to a non-robust model ahead of time, they do have access to the global model throughout the training procedure. We introduce, **ARU-Extract (ARU-E), a method for a small group of colluding clients to extract the non-robust model from the robust global model to then perform ARU**. In Figure 2, (a) multiple ARU-E clients initially receive a copy of the robust model during FAT. For a set number of rounds, (b) the ARU-E

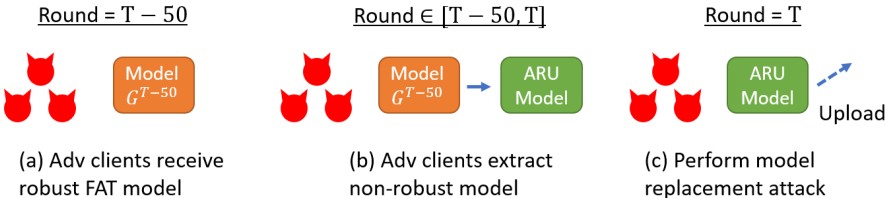

(a) Adv clients receive robust FAT model

(b) Adv clients extract non-robust model

(c) Perform model replacement attack

Figure 2: ARU-E procedure to extract non-robust model from the global FAT model, multiple adversarial clients jointly weaken the FAT model and perform a model replacement attack.

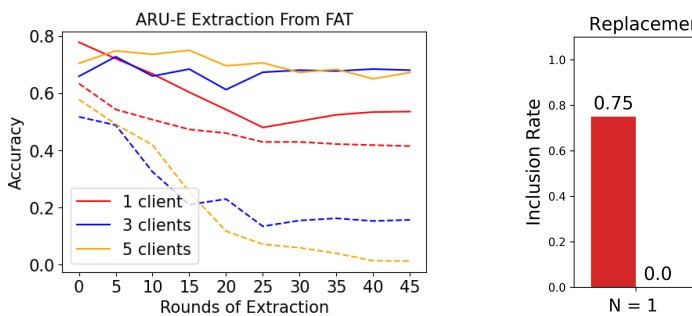

(a) ARU-E for different number of backdoor clients. Solid lines indicate test accuracy and dashed lines show accuracy against evasion attacks (Adv. Acc.).

(b) ARU-E update inclusion probability against trimmed mean and median defenses.

Figure 3: ARU-E method analysis on the CIFAR-10 data set for varying number of backdoor clients.

clients jointly update the FAT model on perturbed data. Here, data points are perturbed using PGD, and an incorrect label is assigned based on how the new model classifies the perturbed data. Such manipulation and training on the ARU-E clients' data induces catastrophic forgetting [24], where the robustness of the global model is forgotten via overfitting on the manipulated data. Finally, (c) the adversaries jointly perform the model replacement ARU attack with the extracted model.

The performance of ARU-E is shown in Table 1. Here, 5 clients out of 40 jointly perform ARU-E for 50 rounds. High test accuracy as well as lower adversarial robustness accuracy indicates the success of the extraction method. We see that the robustness of the ARU-E model is even lower when compared to injecting the FedAvg model by ARU. The inclusion of perturbed data for the ARU-E pushes the decision boundary to be even more feeble than FedAvg. Figure 3a shows that the catastrophic forgetting of robustness occurs very quickly with a limited number of training rounds amongst the relatively small group of 5 adversaries, although test and adversarial accuracy is worse with 1 and 3 adversaries. Furthermore, the extraction procedure of ARU-E degrades test accuracy, and future iterations of it require alterations to keep the the test accuracy high.

# 5 Defense Against ARU and Discussion

Robust aggregation schemes, such as trimmed-mean and median methods, are the primary defenses against poisoning and backdoor attacks [25]. In general, such defenses aim to discard updates from clients that stray too far from the average update, as adversarial updates tend to show high deviation from benign updates. In this section, we first observe the effect of robust aggregation schemes against the ARU attack, and then perform an extended discussion on how the ARU attack may be improved.

## 5.1 Performance of ARU Against Robust Aggregation Schemes

Both the trimmed-mean and median methods sort the parameters of all updates (values or gradients) in ascending order. The median method selects the median value for each individual parameter, while the trimmed-mean method discards $\beta \in [0, 1]$ proportion of largest and smallest values. We set $\beta = 0.15$. The ARU-E method (5 attacking clients out of 40) is analyzed in Table 2. Overall, the use of robust-aggregation schemes significantly reduces the success of the ARU-E attack, more so for the

| Dataset | Metric | Trimmed Mean | Median |
|---------|--------|--------------|--------|
| CIFAR10 | Test Acc. | 0.752 (0.06) | 0.785 (0.05) |
|  | Adv. Acc. | 0.209 (0.04) | 0.621 (0.07) |
| CIFAR100 | Test Acc. | 0.433 (0.06) | 0.485 (0.05) |
|  | Adv. Acc. | 0.293 (0.05) | 0.427 (0.06) |

Table 2: Performance of robust aggregation defense against ARU-E.

median method as only a single update value for each parameter is selected. As seen in Figure 3b, the updates from the adversarial clients are often excluded against the median defense, and less so for the trimmed mean defense. This problem is slightly mitigated as the number of adversaries increases, as the boosting required in the model replacement phase is spread out across more clients.

## 5.2 Discussion: The Next Steps of ARU

As seen in Figure 3b, spreading the model replacement attack across multiple ARU clients reduces the effectiveness of the robust aggregation defense. Also, as the FAT and ARU-E models have similar performance regarding benign test accuracy, the two models are comparatively not too far off compared to label swapping or boosting poisoning attacks [26], leading to higher inclusion rates against trimmed mean. We next aim to spread the ARU attack through multiple rounds of aggregation, further decreasing the required boosting per client per round to reduce the effectiveness of the defense. This allows us to go beyond the model injection attack and utilize other gradual backdoor methods. However, spreading the model replacement attack across multiple rounds of training induces benign clients to begin to contribute more to training, as the model is further from convergence (i.e., removing the assumption made between Equation 4 and 5): defense effectiveness may further be reduced, as benign clients no longer have near zero updates, leading to the adversarial uploads standing out less as outliers.

The robust aggregation schemes employed in federated learning, specifically the trimmed-mean and median methods, exhibit certain limitations and drawbacks. Both methods entail an overhead as they necessitate the sorting of every individual weight parameter in the neural network in either ascending or descending order. While the median method demonstrates robust defensive capabilities in experimental settings, it is not conducive to computing the weighted average during the aggregation phase of federated learning, making it less effective when dealing with data that is distributed in a highly non-i.i.d. fashion across clients. On the other hand, the trimmed mean method can be adapted to incorporate weighted averages during aggregation but incurs a higher computational overhead and weakness against ARU in comparison. Future attack approaches will be effective against more advanced defenses [27] that often are more sensitive to amplification and update vector direction. Consequently, the choice of robust aggregation method involves significant trade-offs among factors such as overhead, defensive efficacy, and performance on the original classification task.

Moving forward, we aim to enhance ARU against robust aggregation schemes, and utilize the ideas of spreading the model replacement procedure across multiple rounds. After an effective attack is developed, we will turn our attention to addressing the issues of the defenses against ARU.

## 6 Conclusion

The landscape of federated learning presents unique challenges, particularly the vulnerability to a multitude of adversaries due to the participation of numerous clients in both training and testing phases. In response to the growing threat of test-time evasion attacks, federated adversarial training has emerged as a promising defense mechanism. Our contribution, Adversarial Robustness Unhardening (ARU), represents a novel train-time counterpart to test-time evasion attacks, discreetly undermining the robustness of models trained using federated adversarial training. Through comprehensive experimentation across diverse scenarios and data sets, we have demonstrated ARU's effectiveness in reducing model robustness. Furthermore, our evaluation of ARU against robust aggregation schemes sheds light on both the potential mitigation and limitations of the ARU attack. These findings lead to a fruitful discussion on the vulnerabilities inherent in robust aggregation schemes and offer insights into enhancing the ARU attack, as well as strategies to fortify the weaknesses of existing aggregation schemes. Our work contributes to the ongoing efforts to secure federated learning against adversarial threats while advancing the understanding of its intricate dynamics.

## Acknowledgments and Disclosure of Funding

This research was partially supported by the Nicholas Minnici (E '59) Dean's Graduate Fellowship in Electrical and Computer Engineering and by the CyLab Security and Privacy Institute, both in affiliation with Carnegie Mellon University. Special thanks to Yixin Yang of Carnegie Mellon University (Electrical and Computer Engineering department, class of 2024) who has helped compile and organize the code and implementation of this paper. The repo she has organized is found at `https://github.com/YixinYang69/newFedEM`.

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
