# OpenReview forum: "Adversarial Robustness Unhardening via Backdoor Attacks in Federated Learning"
_NeurIPS.cc/2023/Workshop/BUGS — NeurIPS 2023 BUGS Poster_

### Official Review · Reviewer_9Nz5 · 2023-10-24
**Review for Submission19**

**Rating:** 6
**Confidence:** 3

**Review:**

This paper presents a new type of attack with both backdoor and adversarial attacks. The core idea is to break the ongoing adversarial training procedure during FL. It first injects a backdoor into the target model during FL via model replacement with an extracted (nonrobust model).  And the nonrobust model will break the adversarial training of the global model. Experiments show the effectiveness of the proposed method.

Weaknesses:
1. It is not clear whether it is a backdoor attack or an adversarial attack. The idea of replacing the current global model with one of its extracted nonrobust versions is interesting.

2. What is the minimum number of malicious clients that could break the global model, 1 or 5?

3. More advanced and robust aggregation rules should be tested, except trimmed mean and median.

4. FL is still centralized learning as it has a central parameter server.

---

### Official Review · Reviewer_zXfh · 2023-10-27
**This paper introduces ARU attacks that decrease the performance of models trained with federated adversarial training**

**Rating:** 6
**Confidence:** 4

**Review:**

Summary:

This paper proposes a new "Adversarial Robustness Unhardening" (ARU) attack that undermines the robustness of models trained with federated adversarial training. The key idea is coordinating train-time and test-time attacks to reduce model robustness to evasion attacks. The authors introduce the ARU attack concept and an ARU-Extract method to obtain a non-robust model during training. Experiments demonstrate ARU's impact in reducing robustness. The paper also evaluates defenses like robust aggregation, and discusses enhancing ARU against them.

Strengths:
- Novel idea of coordinating train-time and test-time attacks.
- Thorough evaluation of ARU's impact on model robustness.
- Interesting concept of extracting non-robust model during training.

Weaknesses:
- Provide the difference between ARU and ARU-E.
- Any other backdoor attacks that can be used for ARU instead of model replacement, since model replacement can be easily detected by the server (Median defense).

---

### Decision · Program_Chairs · 2023-10-28

**Decision:**

Accept (Poster)

**Comment:**

Thanks for submitting to BUGS workshop! Both reviewers recommend acceptance.